# CONSENSUS CLUSTERING WITH UNSUPERVISED REPRESENTATION LEARNING

## ABSTRACT

Recent advances in deep clustering and unsupervised representation learning are based on the idea that different views of an input image (generated through data augmentation techniques) must either be closer in the representation space, or have a similar cluster assignment. In this work, we leverage this idea together with ensemble learning to perform clustering and representation learning. Ensemble learning is widely used in the supervised learning setting but has not yet been practical in deep clustering. Previous works on ensemble learning for clustering neither work on the feature space nor learn features. We propose a novel ensemble learning algorithm dubbed Consensus Clustering with Unsupervised Representation Learning (ConCURL) which learns representations by creating a consensus on multiple clustering outputs. Specifically, we generate a cluster ensemble using random transformations on the embedding space, and define a consensus loss function that measures the disagreement among the constituents of the ensemble. Thus, diverse ensembles minimize this loss function in a synergistic way, which leads to better representations that work with all cluster ensemble constituents. Our proposed method ConCURL is easy to implement and integrate into any representation learning or deep clustering block. ConCURL outperforms all state of the art methods on various computer vision datasets. Specifically, we beat the closest state of the art method by 5.9 percent on the ImageNet-10 dataset, and by 18 percent on the ImageNet-Dogs dataset in terms of clustering accuracy. We further shed some light on the under-studied overfitting issue in clustering and show that our method does not overfit as much as existing methods, and thereby generalizes better for new data samples.

## 1 INTRODUCTION

Supervised learning algorithms have shown great progress recently, but generally require a lot of labeled data. However, in many domains (e.g., advertising, social platforms, etc.), most of the available data are not labeled and manually labeling it is a very labor, time, and cost intensive task (Xiao et al., 2015; Deshmukh, 2019; Mintz et al., 2009; Blum & Mitchell, 1998). On the other hand, clustering algorithms do not need labeled data to group similar data points into clusters. Some popular clustering algorithms include k-means, hierarchical clustering, DBSCAN (Ester et al., 1996), spectral clustering, etc., and the usefulness of each algorithm varies with the application. In this work, we deal with the clustering of images. Traditional clustering approaches focus on hand crafted features on which out of the box clustering algorithms are applied. However, hand crafted features may not be optimal, and are not scalable to large scale real word datasets (Wu et al., 2019). Advancements in deep learning techniques have enabled end-to-end learning of rich representations for supervised learning. On the other hand, simultaneously learning the feature spaces while clustering leads to degenerate solutions, which until recently limited end to end implementations of clustering with representation learning approaches (Caron et al., 2018). Recent deep clustering works take several approaches to address this issue such as alternating pseudo cluster assignments and pseudo supervised training, comparing the predictions with their own high confidence assignments (Caron et al., 2018; Asano et al., 2019; Xie et al., 2016; Wu et al., 2019), and maximizing mutual information between predictions of positive pairs (Ji et al., 2019). Although these methods show impressive performance on challenging datasets, we believe taking advantage of rich ideas from ensemble learning for clustering with representation learning will enhance the performance of deep clustering methods.

Ensemble learning methods train a variety of learners and build a meta learner by combining the predictions of individual learners (Dietterich, 2000; Breiman, 1996; Freund et al., 1996). In practice they have been heavily used in supervised learning setting. Ensemble learning methods have also found their place in clustering i.e. knowledge reuse framework (Strehl & Ghosh, 2002) where a consensus algorithm is applied on constituent cluster partitions to generate an updated partition that clusters the data better than any component partition individually. However, the knowledge reuse framework and much of the consensus clustering literature that followed (Fern & Brodley, 2003; Fred & Jain, 2005; Topchy et al., 2005) do not make use of the underlying features used to generate the ensemble. We propose the use of consensus clustering as a way to extend ensemble methods to unsupervised representation learning. In particular, we define a 'disagreement' measure among the constituents of the ensemble. The key motivation for this is that the diversity of the ensemble drives the minimization of the disagreement measure in a synergistic way, thereby leading to better representations. We propose Consensus Clustering with Unsupervised Representation Learning (ConCURL ) and following are our main contributions:

1. A novel ensemble learning algorithm which learns representations by creating a consensus on multiple clustering outputs generated by applying random transformations on the embeddings.

2. Our method outperforms the current state of the art clustering algorithms on popular computer vision datasets based on clustering metrics ( A.4).

3. Even though there is no labeled data available while learning representations, clustering may still be prone to be overfitting to the "training data." As stated in Bubeck & Von Luxburg (2007), in clustering, we generally assume that the finite data set has been sampled from some underlying space and the goal is to find the true approximate partition of the underlying space rather than the best partition in a given finite data set. Hence, to check generalizability of the method proposed we also evaluate our models on the "test data" - data that was not available during training/representation learning. Our method is more generalizable compared to state of the art methods (i.e. it outperforms the other algorithms when evaluated on the test set).

## 2 RELATED WORK

Clustering is a ubiquitous task and it has been actively used in many different scientific and practical pursuits such as detecting genes from microarray data (Frey & Dueck, 2007), clustering faces (Rodriguez & Laio, 2014), and segmentation in medical imaging to support diagnosis (Masulli & Schenone, 1999). We refer interested readers to these excellent sources for a survey of these uses Jain et al. (1999); Liao (2005); Xu & Wunsch (2005); Nugent & Meila (2010).

**Clustering with Deep Learning:** In their influential work, Caron et al. (2018) show that it is possible to train deep convolutional neural networks with pseudo labels that are generated by a clustering algorithm (DeepCluster). More precisely, in DeepCluster, previous versions of representations are used to assign pseudo labels to the data using an out of the box clustering algorithm such as k-means. These pseudo labels are used to improve the learned representation of the data by minimizing a supervised loss. Along the same lines, several more methods have been proposed. For example, Gaussian ATtention network for image clustering (GATCluster) (Niu et al., 2020) comprises four self-learning tasks with the constraints of transformation invariance, separability maximization, entropy analysis and attention mapping. Training is performed in two distinct steps, similar to Caron et al. (2018) where the first step is to compute pseudo targets for a large batch of data and the second step is to train the model in a supervised way using the pseudo targets. Both DeepCluster and GATCluster use k-means to generate pseudo labels which may not scale well. Wu et al. (2019) propose Deep Comprehensive Correlation Mining (DCCM), where discriminative features are learned by taking advantage of the correlations of the data using pseudo-label supervision and triplet mutual information among features. However, DCCM may be susceptible to trivial solutions (Niu et al., 2020). Invariant Information Clustering (IIC) (Ji et al., 2019) maximizes mutual information between the class assignments of two different views of the same image (paired samples) in order to learn representations that preserve what is common between the views while discarding instance specific details. Ji et al. (2019) argue that the presence of an entropy term in mutual information plays an important role in avoiding the degenerate solutions. However a large batch size is needed

for the computation of mutual information in IIC, which may not be scalable for larger image sizes which are common in popular datasets (Ji et al., 2019; Niu et al., 2020). Huang et al. (2020) extend the traditional maximal margin clustering idea to the deep learning paradigm, by learning the most semantically plausible clustering through minimizing a proposed partition uncertainty index. Their algorithm PICA uses a stochastic version of the index, thereby facilitating mini-batch training. PICA fails to assign a sample-correct cluster when that sample either has high foreground or background similarity to samples in other clusters. SCAN (Van Gansbeke et al., 2020) uses the learnt representations from a pretext task to find semantically closest images to the given image using nearest neighbors. SCAN achieves state-of-the-art results on CIFAR-10, CIFAR-100, STL-10, however using such priors from existing pretext tasks deviates from the main idea of our paper of end-to-end learning. We believe however that we can benefit by using SCAN on top of the features learnt using our algorithm to improve the clustering accuracy. Our proposed approach ConCURL is scalable to large datasets, does not suffer from trivial solutions and shows superior performance on a challenging set of image data sets. As shown in the experimental results, our proposed method also generalizes well to data points that were not available during training, when compared to the above approaches.

**Self-supervised Representation Learning:** Self-supervised learning is a sub-field of unsupervised learning in which the main goal is to learn general purpose representations by exploiting user-defined sub-tasks such as the relationship between different views of the same data. Although self-supervised learning methods show impressive performance on a variety of problems, it is not clear whether learned representations are good for clustering. There are many different flavors of self supervised learning such as Instance Recognition tasks (Wu et al., 2018), contrastive techniques (Chen et al., 2020), etc. In instance recognition tasks, each image is considered as its own category so that the learnt embeddings are well separated. Zhuang et al. (2019) propose a novel Local Aggregation method based on non-parametric instance discrimination tasks. They use a robust clustering objective (using multiple runs of k-means) to move statistically similar data points closer in the representation space and dissimilar data points further away. However, instance recognition tasks usually require a large memory bank which is memory intensive. In contrastive learning (Tian et al., 2019; He et al., 2020; Hénaff et al., 2019; Hjelm et al., 2018; Chen et al., 2020), representations are learned by maximizing agreement between different augmented views of the same data example (known as positive pairs) and minimizing agreement between augmented views of different examples (known as negative pairs). SimCLR (Chen et al., 2020) achieves state-of-the-art-results without specialized architectures or a memory bank of negative pairs (usually required by contrastive learning techniques). However, it still requires negative examples and, as it applies to instance classification, it has to compare every pair of images. These issues are addressed by Bootstrap your own latent (BYOL) (Grill et al., 2020) and Swapping Assignments between multiple Views (SwAV) (Caron et al., 2020). BYOL does not require negative examples and SwAV does not need to compare every pair of images. For the main study in this paper we use BYOL as a representation learning block and adapt the soft clustering loss which was used in SwAV to learn prototypes, thereby addressing both the issues of negative samples and need for comparing every pair of images. Note that our proposed algorithm can use any representation learning block like SimCLR, BYOL, & SwAV and can also use other soft clustering loss formulations.

**Consensus Clustering:** Strehl & Ghosh (2002) propose a knowledge reuse framework; to build an unsupervised ensemble by using several distinct clusterings of the same data and assuming that the underlying features that are used to compute the clustering are not available and fixed. Fern & Brodley (2003) build on the cluster ensemble framework based on random projections. In this framework, Fern & Brodley (2003) show that a single run of clustering (random projection + Expectation Maximization) is highly unstable. They perform multiple runs of clustering and compute an aggregated similarity matrix which is used to cluster the data using an agglomerative clustering algorithm. Fred & Jain (2005) propose a voting approach to map the cluster assignments in the ensemble into a new similarity measure between clusterings. The co-association matrix thus formed can be used with any clustering algorithm to result in a new data partition. Weighted cluster ensembles (Domeniconi & Al-Razgan, 2009) uses Locally Adaptive Clustering (LAC) (Domeniconi et al., 2004) as a base algorithm which assigns weights to each feature of the sample according to the local variance of data along each dimension. Authors use LAC with different hyperparameters and propose three different partitioning algorithms to generate consensus among LAC clusterings. Huang et al. (2017) argues that most ensemble based methods treat all base clusterings equally and a few which weigh the base clusterings give those weights globally. Authors propose an ensemble clustering approach based

on cluster uncertainty (calculated by entropic criterion) and a local weighting strategy. There are also survey papers which have in depth analysis and details of various methods used in consensus clustering Vega-Pons & Ruiz-Shulcloper (2011); Ghosh & Acharya (2011). It is not clear how any of these methods can be adapted when one needs to do representation learning along with clustering. It is also not clear if one can come up with end-to-end learning architecture with any of the above methods. In contrast, our proposed consensus clustering method ConCURL can easily be integrated into any deep learning architecture for clustering and trained end-to-end.

## 3 PROPOSED METHOD

Given a set of observations $\mathcal{X} = \{x_i\}_{i=1}^N$, the goal is to learn a representation $f$, cluster representations $C$ (called as prototypes henceforth) and partition $N$ observations into $K$ disjoint clusters (the value of $K$ is assumed to be known). To check the generalization of the algorithm, the final goal is to partition previously unseen $N_T$ observations $\mathcal{X}_\mathcal{T} = \{x_i\}_{i=N+1}^{N+N_T}$ into $K$ disjoint clusters given learnt representation $f$ and prototypes $C$.

### 3.1 ALGORITHM

We propose a novel consensus loss to achieve the objective. Along with the novel consensus clustering loss, the loss function comprises of two more loss terms, namely a representation learning loss, a soft clustering loss. The three loss terms share a common backbone (such as ResNet) and have different loss specific layers after the representation layer of the backbone. Given an input batch $\mathcal{X}_b \subset \mathcal{X}$ of size $B$, we use different data augmentations in the image space, such as random horizontal flip, etc. (more details in A.3) to generate two augmented views $\mathcal{X}_b^1, \mathcal{X}_b^2$ of the input batch. $\mathbf{f}_b^1, \mathbf{f}_b^2$ are representations that are corresponding to the two views of the input batch. The superscript is used to identify the view of the input image. It's usage will be made clear from context, and ignored otherwise. Our algorithm is mini-batch based, and end-to-end.

#### 3.1.1 UNSUPERVISED REPRESENTATION LEARNING

The first loss term is an unsupervised representation learning loss based on a pretext task such as BYOL (Grill et al., 2020), SimCLR (Chen et al., 2020), etc. In this work, we use BYOL which tries to minimize the distance between the online and target network embeddings of the two augmented views $\mathcal{X}_b^1, \mathcal{X}_b^2$. We denote this loss by $L_1$. Optimizing for loss $L_1$ alone generally results in good representations but may not be suitable for a clustering task.

#### 3.1.2 SOFT CLUSTERING

For the second loss term, we use a soft clustering objective. In this work, we follow the framework presented in Caron et al. (2020), which is a centroid based technique and aims to maintain consistency between the clusterings of the augmented views $\mathcal{X}_b^1$ and $\mathcal{X}_b^2$. We store a set of randomly initialized prototypes $C = \{\mathbf{c}_1, \cdots, \mathbf{c}_K\} \in \mathbb{R}^{d \times K}$, where $K$ is the number of clusters, and $d$ is the dimension of the prototypes.

We use a two layer multi-layer perceptron $g$ to project the features $\mathbf{f}^1$ and $\mathbf{f}^2$ to a lower dimensional space (of size $d$). This technique was shown to be useful in improving the representations of the layer before the MLP (Chen et al., 2020) and thereafter used in Grill et al. (2020) and Caron et al. (2020). The output of this MLP (referred to as embeddings) is denoted using $Z^1 = \{\mathbf{z}_1^1, \ldots, \mathbf{z}_B^1\}$ and $Z^2 = \{\mathbf{z}_1^2, \ldots, \mathbf{z}_B^2\}$ for view 1 and view 2 respectively.

Soft clustering approaches based on centroids/prototypes often requires one to compute a measure of similarity between the image embeddings and the prototypes (Xie et al., 2016; Caron et al., 2020). We compute the probability of assigning a cluster $j$ to image $i$ using the normalized vectors $\bar{\mathbf{z}}_i^1 = \frac{\mathbf{z}_i^1}{||\mathbf{z}_i^1||}, \bar{\mathbf{z}}_i^2 = \frac{\mathbf{z}_i^2}{||\mathbf{z_i}^2||}$ and $\bar{\mathbf{c}}_j = \frac{\mathbf{c_j}}{||\mathbf{c_j}||}$ as

$$\mathbf{p}_{i,j}^1 = \frac{\exp(\frac{1}{\tau}\langle \bar{\mathbf{z}}_i^1, \bar{\mathbf{c}}_j \rangle)}{\sum_{j'} \exp(\frac{1}{\tau}\langle \bar{\mathbf{z}}_i^1, \bar{\mathbf{c}}_{j'} \rangle)}, \quad \mathbf{p}_{i,j}^2 = \frac{\exp(\frac{1}{\tau}\langle \bar{\mathbf{z}}_i^2, \bar{\mathbf{c}}_j \rangle)}{\sum_{j'} \exp(\frac{1}{\tau}\langle \bar{\mathbf{z}}_i^2, \bar{\mathbf{c}}_{j'} \rangle)}. \tag{1}$$

We concisely write $\mathbf{p}_i^1 = \{\mathbf{p}_{i,j}^1\}_{j=1}^K$ and $\mathbf{p}_i^2 = \{\mathbf{p}_{i,j}^2\}_{j=1}^K$. Here, $\tau$ is a temperature parameter. Note that, we use $\mathbf{p}_i$ to denote the predicted cluster assignment probabilities for image $i$ (when not referring to a particular view), and a shorthand $\mathbf{p}$ is used when $i$ is clear from context.

The idea of predicting assignments $\mathbf{p}$, and then comparing them with high-confidence estimates $\mathbf{q}$ (referred to as codes henceforth) of the predictions is not new (Xie et al., 2016). While Xie et al. (2016) uses pretrained features (from autoencoder) to compute the predicted assignments and the codes, doing this in an end to end unsupervised manner might lead to degenerate solutions. Asano et al. (2019) avoids such degenerate solutions by enforcing an equi-partition constraint (the prototypes equally partition the data) during code computation. Caron et al. (2020) follow the same formulation but compute the codes in an online manner for each mini-batch. The assignment codes are computed by solving the following optimization problem

$$Q^1 = \underset{Q \in \mathcal{Q}}{\arg\max} \operatorname{Tr}(Q^T C^T Z^1) + \epsilon H(Q) \text{ and } Q^2 = \underset{Q \in \mathcal{Q}}{\arg\max} \operatorname{Tr}(Q^T C^T Z^2) + \epsilon H(Q), \quad (2)$$

where $Q = \{\mathbf{q}_1, \ldots, \mathbf{q}_B\} \in \mathbb{R}_+^{K \times B}$, $\mathcal{Q}$ is the transportation polytope defined by

$$\mathcal{Q} = \{\mathbf{Q} \in \mathbb{R}_+^{K \times B} | \mathbf{Q}\mathbf{1}_B = \frac{1}{K}\mathbf{1}_K, \mathbf{Q}^T\mathbf{1}_K = \frac{1}{B}\mathbf{1}_B\}$$

$\mathbf{1}_K$ is a vector of ones of dimension $K$ and $H(Q) = -\sum_{i,j} Q_{i,j} \log Q_{i,j}$. The above optimization is computed using a fast version of the Sinkhorn-Knopp algorithm (Cuturi, 2013) as described in Caron et al. (2020).

After computing the codes $Q^1$ and $Q^2$, we compute the loss using the probabilities $\mathbf{p}_{ij}$ and the assigned codes $\mathbf{q}_{ij}$ by comparing the probabilities of view 1 with the assigned codes of view 2 and vice versa, given as

$$L_{2,1} = -\frac{1}{2B}\sum_{i=1}^{B}\sum_{j=1}^{K}\mathbf{q}_{ij}^2 \log \mathbf{p}_{ij}^1, \quad L_{2,2} = -\frac{1}{2B}\sum_{i=1}^{B}\sum_{j=1}^{K}\mathbf{q}_{ij}^1 \log \mathbf{p}_{ij}^2, \quad L_2 = L_{2,1} + L_{2,2} \quad (3)$$

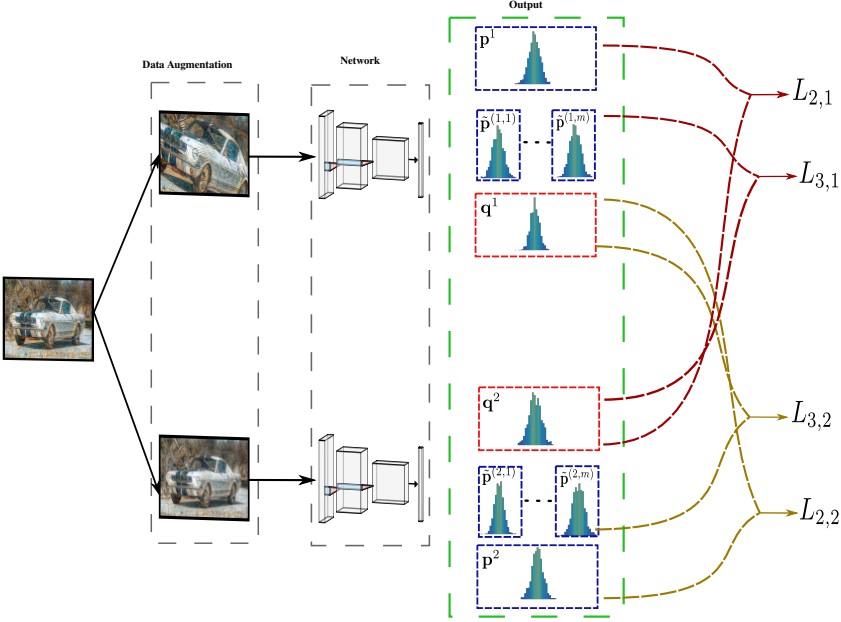

Figure 1: The resulting ensemble of clusterings $\{\mathbf{p}^i, \mathbf{q}^i, \{\tilde{\mathbf{p}}^{(i,m)}\}_m\}_{i=\{1,2\}}$ is represented in the green block. $L_{2,1}, L_{2,2}$ represent soft clustering loss as in eqn. (3), $L_{3,1}, L_{3,2}$ represent consensus loss as in eqn. (4)

### 3.1.3 Consensus Clustering

The third loss term is the proposed consensus clustering loss. We generate a cluster ensemble by first performing transformations on the embeddings $Z^1$, $Z^2$ and the prototypes $C$. At the beginning of the algorithm we randomly initialize $M$ such transformations and fix them throughout training. Suppose using a particular random transformation (a randomly generated matrix $A$), we get $\tilde{\mathbf{z}} = A\mathbf{z}$, $\tilde{\mathbf{c}} = A\mathbf{c}$. We then compute the softmax probabilities $\tilde{\mathbf{p}}_{ij}$ using the normalized vectors $\tilde{\mathbf{z}}/||\tilde{\mathbf{z}}||$ and $\tilde{\mathbf{c}}/||\tilde{\mathbf{c}}||$. Repeating this with the $M$ transformations results in $M$ predicted cluster assignment probabilities for each view. When the network is untrained, the embeddings $\mathbf{z}$ are random and applying the random transformations followed by computing the predicted cluster assignments leads to a diverse set of soft cluster assignments.

There can be many clusterings for any given latent space (either by randomization introduced in the space or different clustering algorithms). We propose to use consensus clustering so that we can find a latent space that maximizes performance on many different clusterings. To make the method effective, we need more diversity in the clusterings among each component of the ensemble at the start and ideally the diversity should decrease with training and the clustering performance should get better. Creating such a diverse ensemble leads to better representations and better clusters.

To compute the consensus loss, once the probabilities $\tilde{\mathbf{p}}_{ij}$ are computed, we compare the codes generated using (2) of view 1 with the $\tilde{\mathbf{p}}$ of view 2 and vice versa, given as

$$L_{31} = -\frac{1}{2BM} \sum_{i=1}^{B} \sum_{m=1}^{M} \sum_{j=1}^{K} \mathbf{q}_{ij}^2 \log \tilde{\mathbf{p}}_{ij}^{(1,m)}, \quad L_{32} = -\frac{1}{2BM} \sum_{i=1}^{B} \sum_{m=1}^{M} \sum_{j=1}^{K} \mathbf{q}_{ij}^1 \log \tilde{\mathbf{p}}_{ij}^{(2,m)} \quad (4)$$

$$L_3 = L_{31} + L_{32} \quad (5)$$

The architecture for computing the ensemble and the computation of consensus loss is shown in Fig. (1). The final loss that we sought to minimize is the combination of the losses $L_1, L_2, L_3$

$$L_{\text{total}} = \alpha L_1 + \beta L_2 + \gamma L_3. \quad (6)$$

where $\alpha, \beta, \gamma$ are non-negative constants. $L_1$ minimizes the distance between the embeddings of different views of the input batch, $L_2$ maintains consistency between clusterings of two augmented views of the input batch, and $L_3$ maintains consistency between clusterings of the randomly transformed embeddings. During inference, to compute a clustering of the input images, we use the computed assignments $\{\mathbf{q}_i\}_{i=1}^{N}$ and assign the cluster index as $c_i = \arg\max_k \mathbf{q}_{ik}$ for the $i^{\text{th}}$ datapoint. The summary of the algorithm is presented in A.2.

### 3.2 Choice of Feature Transformations

Table 1: Different ways to generate ensembles

| Data Representation | Clustering algorithms |
| --- | --- |
| Different data preprocessing techniques | Multiple clustering algorithms (k-means, GMM, etc) |
| Subsets of features | Same algorithm with different parameters or initializations |
| Different transformations of the features | Combination of multiple clustering and different parameters or initializations |

Fred & Jain (2005) discuss different ways to generate an ensemble of clusterings which are tabulated in Table 1. In our proposed algorithm, we focus on choosing of data representation to generate cluster ensembles.

Using different clustering methods (or multiple runs from different initializations) showed success when the goal is to aggregate the cluster partitions of the ensemble into a final partition. However,

since our goal is to learn representations simultaneously using back-propagation, it is natural to use a clustering algorithm in which we can update its parameters via back-propagation. It is in-fact possible to use clustering algorithms like IIC (Ji et al., 2019) that enable parameter updates, however, if multiple such algorithms are used to generate the ensemble, the backward computation graph consumes an enormous amount of memory. For large datasets (and large architectures), such techniques may not be practical. However, by using feature transformations, the amount of such memory overhead is small.

By fixing a stable clustering algorithm, we can generate arbitrarily large ensembles by applying different transformations on the embeddings. Random projections were successfully used in Consensus Clustering previously (Fern & Brodley, 2003). By generating ensembles using random projections, we have control over the amount of diversity we can induce into the framework, by varying the dimension of the random projection. In addition to Random Projections, we also used diagonal transformations (Hsu et al., 2018) where different components of the representation vector are scaled differently. Hsu et al. (2018) illustrate that such scaling enables a diverse set of clusterings which is helpful for their meta learning task.

## 4 EXPERIMENTS

We evaluated our algorithm and compared against existing work on nine popular image datasets with a mix of high and medium resolution datasets: ImageNet-10, ImageNet-Dogs, STL10, CIFAR-10, CIFAR100-20, CUB, Caltech-101, AwA2 and Intel Image classification. The dataset summary is given in Table 6 (see A.1). The resolution column shows the size to which we resized the images in our algorithm. To the best of our knowledge some of these datasets (Intel, Caltech101, CUB, and AwA2) are being used for systematically evaluating clustering results for the first time (Section A.5). In our comparison, we considered some recent state-of-the-art methods that directly solve for a clustering objective in an end-to-end fashion from random initialization, and do not use any prior information (nearest neighbors for example) derived through other pretext tasks. The metrics for evaluation are explained in A.4.

### 4.1 RESULTS

In Table 2, we show the best accuracy we get for the proposed method ConCURL and compare it against the best accuracy achieved by state-of-the-art methods. For ImageNet-10 and ImageNet-Dogs, we trained using the train split and evaluated on the train split. For STL10, CIFAR10 and CIFAR-100, similar to earlier approaches (Huang et al., 2020; Niu et al., 2020) we used both train and test splits for training and evaluation. Note that PICA also uses unlabelled data split of 100k points in STL10. In all datasets ConCURL outperforms all state-of-the-art algorithms. In the ImageNet-10 dataset, ConCURL is better by 5.9% and 21% compared to PICA and GATCluster respectively in terms of clustering accuracy. ConCURL is better than PICA by more than 9% in terms of NMI and by more than 11% in terms of ARI in the ImageNet-10 dataset. We beat the next-best DCCM method by 18% in ImageNet-Dogs. We used the popular Iman and Daveport Test to calculate the p-value for statistical significance test (Garcia & Herrera, 2008). Our proposed method ConCURL was consistently ranked at the top for all three metrics ACC, ARI and NMI with p-value of $1.34 \times 10^{-4}$, $4.57 \times 10^{-5}$ and $1.45 \times 10^{-3}$ respectively. This shows that results presented in the paper are statistically significant.

Table 2: Clustering evaluation metrics

| Datasets | ImageNet-10 | | | ImageNet-Dogs | | | STL10 | | | CIFAR10 | | | CIFAR100-20 | | |
|---|---|---|---|---|---|---|---|---|---|---|---|---|---|---|---|
| Methods\Metrics | Acc | NMI | ARI | Acc | NMI | ARI | Acc | NMI | ARI | Acc | NMI | ARI | Acc | NMI | ARI |
| DCCM | 0.710 | 0.608 | 0.555 | 0.383 | 0.321 | 0.182 | 0.482 | 0.376 | 0.262 | 0.623 | 0.496 | 0.408 | 0.327 | 0.285 | 0.173 |
| GATCluster | 0.762 | 0.609 | 0.572 | 0.333 | 0.322 | 0.200 | 0.583 | 0.446 | 0.363 | 0.610 | 0.475 | 0.402 | 0.281 | 0.215 | 0.116 |
| PICA | 0.870 | 0.802 | 0.761 | 0.352 | 0.352 | 0.201 | 0.713 | 0.611 | 0.531 | 0.696 | 0.591 | 0.512 | 0.337 | 0.310 | 0.171 |
| ConCURL | **0.922** | **0.877** | **0.852** | **0.452** | **0.447** | **0.288** | **0.728** | **0.634** | **0.554** | **0.720** | **0.608** | **0.520** | **0.385** | **0.377** | **0.223** |

We performed additional experiments to compare ConCURL to PICA when trained only on the "train" split for STL-10, CIFAR-10 and CIFAR-100. From Table 3, we observe that ConCURL achieves better clustering on all metrics. From Table 2 and 3, we can observe that increasing the data for training improves the performance of the algorithm. In order to understand the generalization of

the algorithm, in Table 4, we show results for the case when the models were trained only on "train" split and evaluated on "test data" that was not used during training. We can observe that we perform better when evaluated on the test data.

We also perform an ablation study on the affect of the losses $L_1, L_2, L_3$ (see A.6) and observe that the using consensus loss $L_3$ almost always improves accuracy. This shows the importance of consensus loss ($L_3$) and how ensemble learning through proposed consensus helps in achieving better clusters. Even though proposed method outperforms all state-of-the-art methods there are some assumptions that we make which are same as all state-of-the-methods compared. First assumption is that "K - number of clusters" are known and second assumption is that the data sample is equally distributed among "K" clusters. These assumptions are important to make the methodology work because the final layer in our case needs to know the number of clusters so that we can evaluate it later for fair comparison with other methods. Also, the fast Sinkhorn-Knopp algorithm used to solve Eq. 2 may not be optimal if the data samples are not equally distributed among "K" clusters.

Table 3: Clustering evaluation metrics: Models trained using only train split, and evaluation on train split

| Datasets | STL10 | | | CIFAR10 | | | CIFAR100-20 | | |
|---|---|---|---|---|---|---|---|---|---|
| Methods\Metrics | Acc | NMI | ARI | Acc | NMI | ARI | Acc | NMI | ARI |
| PICA | 0.494 | 0.424 | 0.297 | 0.559 | 0.510 | 0.402 | 0.295 | 0.273 | 0.138 |
| ConCURL | 0.623 | 0.514 | 0.428 | 0.705 | 0.545 | 0.507 | 0.366 | 0.351 | 0.195 |

Table 4: Clustering evaluation metrics: Models trained using only train split, and evaluation on test split of the data

| Datasets | ImageNet-10 | | | ImageNet-Dogs | | | STL10 | | | CIFAR10 | | | CIFAR100-20 | | |
|---|---|---|---|---|---|---|---|---|---|---|---|---|---|---|---|
| Methods\Metrics | Acc | NMI | ARI | Acc | NMI | ARI | Acc | NMI | ARI | Acc | NMI | ARI | Acc | NMI | ARI |
| PICA [1] | 0.758 | 0.724 | 0.602 | 0.375 | 0.399 | 0.208 | 0.484 | 0.422 | 0.293 | 0.559 | 0.504 | 0.393 | 0.291 | 0.276 | 0.138 |
| ConCURL | 0.864 | 0.840 | 0.770 | 0.455 | 0.477 | 0.274 | 0.611 | 0.498 | 0.410 | 0.693 | 0.527 | 0.488 | 0.363 | 0.354 | 0.193 |

## 4.2 EFFECT OF NUMBER OF RANDOM TRANSFORMATIONS AND EMBEDDING SIZE

In order to study the sensitivity of the algorithm to the random transformations, we performed an ablation study for STL10 trained on the train split (Table 5). Recall that $M$ is number of transformations used in algorithm 1 and $M = 30, 50$ yield good results for the two types of transformations (random projections and diagonal transformations). We can also observe that there isn't a large difference between results obtained using either of the transformations. The results fluctuate with a margin of $\pm 0.06$ and still outperform the other baselines in almost all cases. In the first column of the Table 5, we used diagonal transformations, and varied the number of transformations. The second column of the Table 5 contains results with a fixed dimension of random projection (=64), and varies the number of transformations. The third column of the Table 5 contains results with a fixed number of transformations (=100), and varies the dimension of projection (the original embedding size is 256).

Table 5: Ablations on the number of transformations and dimension of random projections. These results were obtained by training ConCURL on train split of STL10 for 1000 epochs.

| Number of Diagonal Transformations | | | | Number of Random Projections | | | | Embedding Dimension Size | | | |
|---|---|---|---|---|---|---|---|---|---|---|---|
| #Trans\Metrics | Acc | NMI | ARI | #Proj\Metrics | Acc | NMI | ARI | #dim\Metrics | Acc | NMI | ARI |
| 10 | 0.567 | 0.514 | 0.373 | 10 | 0.520 | 0.468 | 0.317 | 64 | 0.533 | 0.503 | 0.342 |
| 30 | 0.534 | 0.481 | 0.338 | 30 | 0.586 | 0.515 | 0.386 | 128 | 0.487 | 0.488 | 0.310 |
| 50 | 0.582 | 0.525 | 0.397 | 50 | 0.562 | 0.512 | 0.377 | 256 | 0.525 | 0.502 | 0.338 |
| 100 | 0.530 | 0.489 | 0.344 | 100 | 0.533 | 0.503 | 0.343 | 512 | 0.584 | 0.499 | 0.375 |
| PICA | 0.494 | 0.424 | 0.297 | PICA | 0.494 | 0.424 | 0.297 | PICA | 0.494 | 0.424 | 0.297 |

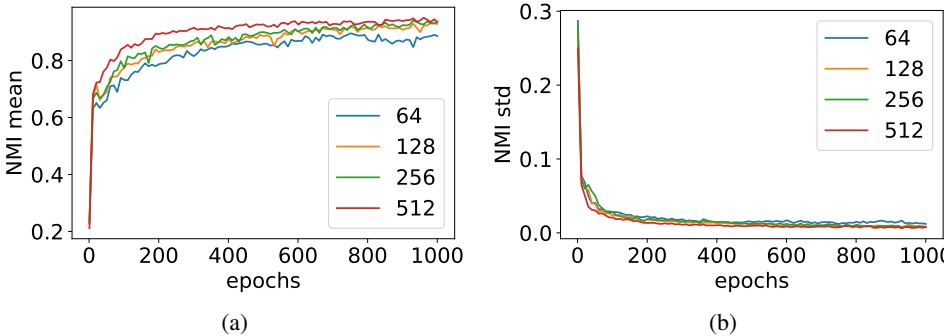

(a)                 (b)

Figure 2: NMI as a way to measure the consensus/diversity in the ensemble. The result are for STL10 trained using the train split.

### 4.3 CONSENSUS AMONG EACH COMPONENT OF THE ENSEMBLE

We also measure the diversity of the ensemble at every epoch to observe the affect of consensus as training progresses. For each component of the ensemble, we use the softmax probability estimates $\mathbf{p}$ and compute cluster assignments by taking an $\arg\max$ on $\mathbf{p}$ of each image. If there are $M$ components in the ensemble, we get $M$ such cluster assignments. We then compute a pairwise NMI (Normalized Mutual Information) (similar analysis to Fern & Brodley (2003)) between every two components of the ensemble, and compute the average and standard deviation of the pairwise NMI across the $\frac{M(M-1)}{2}$ pairs. We observe from Figure 2a that the pairwise NMI increases as training progresses and becomes closer to 1. At the beginning of training, the ensemble is very diverse (small NMI score with a larger standard deviation); and as training progresses, the diversity is reduced (large NMI score with a smaller standard deviation).

## 5 CONCLUSION

In this work, we leverage the ideas in unsupervised representation learning along with ensemble learning to perform clustering. We propose a novel ensemble learning algorithm which learns a representation by creating a consensus on multiple clustering outputs. Our proposed method outperforms all state of the art methods on various computer vision datasets. We also present the issue of overfitting in clustering and show that our method generalizes well on new data samples that were not available during training. This work is one of the first successful applications of ensemble learning in the deep clustering domain. This idea could easily be extended to use other general purpose representation and soft clustering algorithms than the ones used in this paper. A possible extension can be to leverage the knowledge reuse framework of Strehl & Ghosh (2002) and use the clusterings output by the ensemble to compute a better quality partition of the input data. We believe that ensemble learning algorithms could also be effective in increasing robustness in clustering and we are planning to investigate this point further.

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

# A APPENDIX

## A.1 DATASET SUMMARY

The dataset summary is given in Table 6. For ImageNet-10 and ImageNet-Dogs are subsets of Deng et al. (2009). We used the same classes as Chang et al. (2017) for evaluating on these two datasets.

Table 6: Dataset Summary

| Dataset | Classes | Train Data | Test Data | Resolution |
|---|---|---|---|---|
| ImageNet-10 (Deng et al., 2009) | 10 | 13000 | 500 | $224 \times 224$ |
| Imagenet-Dogs (Deng et al., 2009) | 15 | 19500 | 750 | $224 \times 224$ |
| STL-10 (Coates et al., 2011) | 10 | 5000 | 8000 | $96 \times 96$ |
| CIFAR10 (Krizhevsky et al., 2009) | 10 | 50000 | 10000 | $32 \times 32$ |
| CIFAR100-20 (Krizhevsky et al., 2009) | 20 | 50000 | 10000 | $32 \times 32$ |
| CUB (Wah et al., 2011) | 200 | 5994 | 5794 | $224 \times 224$ |
| Caltech-101 (Fei-Fei et al., 2004) | 101 | 7020 | 1657 | $224 \times 224$ |
| Intel | 6 | 14034 | 3000 | $128 \times 128$ |
| AwA2 AwA2(Xian et al., 2018) | 50 | 29865 | 7457 | $224 \times 224$ |

## A.2 ALGORITHM - PSEUDO CODE

The pseudo code for our algorithm is given in Algorithm 1.

---

**Algorithm 1:** Consensus Clustering algorithm (ConCURL )

**Input:** Dataset $\mathcal{X} = \{x_i\}_{i=1}^N$, $K, B, \alpha, \beta, \gamma, M, d$
**Output:** Cluster label $c_i$ of $x_i \in \mathcal{X}$

1 Randomly initialize network parameters $\mathbf{w}$, $K$ prototypes $\mathbf{c}_{1:K}$, $M$ random projection matrices $R_{1:M}$ to dimension $d$ and $e = 0$;
2 **while** $e <$ *total epoch number* **do**
3      **for** $b \in \{1, 2, \ldots, \lfloor \frac{N}{B} \rfloor\}$ **do**
4          Select $B$ samples as $\mathcal{X}_b$ from $\mathcal{X}$ ;
5          Make a forward pass on two views of the input batch $(\mathcal{X}_b^1, \mathcal{X}_b^2)$, and obtain the features $\mathbf{z}_{1:B}^1$, $\mathbf{z}_{1:B}^2$ ;
6          Compute loss $L_1$ which is the representation loss
7          Compute probability of $i^{\text{th}}$ sample belonging to the $j^{\text{th}}$ cluster, $\mathbf{p}_{i,j}$ for both views separately, using normalized $\mathbf{z}$, $\mathbf{c}$ eq ( 1);
8          Compute codes of the current batch $\mathbf{q}$ using eq (2);
9          Compute loss $L_2$ using eq(3);
10          **for** $m \in \{1, 2, \ldots, M\}$ **do**
11              $\tilde{\mathbf{z}}, \tilde{\mathbf{c}} \longleftarrow$ Compute random transformations of $\mathbf{z}, \mathbf{c}$;
12              Compute probability of $i^{\text{th}}$ sample belonging to the $j^{\text{th}}$ cluster, $(\tilde{\mathbf{p}}_{i,j}^{(1,m)}, \tilde{\mathbf{p}}_{i,j}^{(2,m)})$ using normalized $\tilde{\mathbf{z}}, \tilde{\mathbf{c}}$ eq (1);
13          **end**
14          Compute loss $L_3$ using eq (5) ;
15          Compute total loss using eq (6). Update the parameters, and prototypes using gradients
16      **end**
17      $e := e + 1$
18 **end**
19 Make forward pass on all the data and store the features;
20 **foreach** $x_i \in \mathcal{X}$ **do**
21      Compute probability of $i^{\text{th}}$ sample belonging to the $j^{\text{th}}$ cluster, $\mathbf{p}_{i,j}$ using normalized $\mathbf{z}_i, \mathbf{c}_j$ eq (1);
22      Compute codes $\mathbf{q}$ using eq (2);
23      $c_i := \arg\max_k \mathbf{q}_{ik}$;
24 **end**

---

## A.3 IMPLEMENTATION DETAILS

We use a residual network (He et al., 2016) with 34 layers (current state of the art clustering results Huang et al. (2020) also use the same architecture). The MLP projection head consists of a hidden layer of size 2048, followed by batch normalization and ReLU layers, and an output layer of size 256. We use the Adam optimizer (Kingma & Ba, 2014) with a learning rate of 0.0005. We implemented our algorithm using the Pytorch framework and trained our algorithm using a V100 GPU, taking 8 hours to train ImageNet-10 (13,000 training size) with a batch size of 128 for 500 epochs using 1 GPU.

We explain the generation of multiple augmented views that are shown to be very effective in unsupervised learning (Chen et al., 2020). Note that these augmented views are different from the views of a Multi-view image data set such as Schops et al. (2017). Indeed it is possible to use more than two augmented views, but we limit to two for the sake of simplicity. Caron et al. (2020) propose an augmentation technique (Multi-Crop) to use more than two views. In this work, we use the augmentations used in Chen et al. (2020); Grill et al. (2020). We first crop a random patch of the image with scale ranging from 0.08 to 1.0, and resize the cropped patch to 224×224 (128×128 and 96×96 in the case of smaller resolution datasets such as Intel and STL10 respectively). The resulting image was then flipped horizontally with a probability of 0.5. We then apply color transformations, starting by randomly changing the brightness, contrast, saturation and hue with a probability of 0.8. The image was then changed to gray-scale with a probability of 0.2. Then we applied a Gaussian Blur with kernel size (23×23) and a sigma chosen uniformly randomly between 0.1 and 2.0. The probability of applying the Gaussian Blur was 1.0 for view 1 and 0.5 for view 2. During evaluation, we resized the image such that the smaller edge of the image is of size 256 (not required for STL, Intel, CIFAR10, CIFAR100-20), and a center crop is performed with the resolution mentioned in Table 6. The color transformations were computed using Kornia (Riba et al., 2020) which is a differentiable computer vision library for Pytorch.

To compute the random transformations on the embeddings $\mathbf{z}$, we followed two techniques. We used random projections (Bingham & Mannila, 2001) with output dimension $d$, and transformed the embeddings $\mathbf{z}$ to the new space with dimension $d$. We also used diagonal transformation (Hsu et al., 2018) where we multiply $\mathbf{z}$ with a randomly generated diagonal matrix of the same dimension as $\mathbf{z}$. We initialized $M$ random transformations at the beginning and remain fixed throughout the training.

We performed model selection on the hyperparameters of the random transformations on the embedding space such as the number of random transformations $M$ (ranging from 10 to 100) and the dimensionality of the output space if using a random projection (we used [32, 64, 128, 256, 512]). We evaluated the models based on the metrics mentioned in Section A.4 on the data used for training the representations. Note that we fixed the number of prototypes to be equal to the number of ground truth classes. It was shown however that over-clustering leads to better representations (Caron et al., 2020; Ji et al., 2019; Asano et al., 2019) and we can extend our model to include an over-clustering block with a larger set of prototypes (Ji et al., 2019) and alternate the training procedure between the blocks.

### A.4 EVALUATION METRICS

We evaluate our algorithm by computing traditional clustering metrics (Cluster Accuracy, Normalized Mutual Information, and Adjusted Rand Index). Note that for measuring clustering metrics, the usual approach taken in literature is to evaluate the cluster metrics on the train data. Here, we report results on both the train data, as well as test data separately.

**Cluster Accuracy**  The clustering accuracy is computed by first computing a cluster partition of the input data. Once the partitions are computed and cluster indices assigned to each input data point, the linear assignment map is computed using Kuhn-Munkres (Hungarian) algorithm that reassigns the cluster indices to the true labels of the data. Clustering accuracy is then given by

$$ACC = \frac{\sum_{i=1}^{N} \mathbb{1}\{y_{true}(x_i) = c(x_i)\}}{N},$$

where $y_{true}(x_i)$ is a true label of $x_i$ and $c(x_i)$ is the cluster assignment produced by an algorithm (after Hungarian mapping).

**Normalized Mutual Information**  For two clusterings $U, V$, with each containing $|U|, |V|$ clusters respectively, and let $|U_i|$ be the number of samples in cluster $U_i$ of clustering $U$ (similarly for $V$), Mutual Information (MI) is given by

$$MI(U, V) = \sum_{i=1}^{|U|} \sum_{j=1}^{|V|} \frac{|U_i \cap V_i|}{N} \log \frac{N|U_i \cap V_j|}{|U_i||V_j|}$$

where $N$ is the number of data points under consideration. Normalized Mutual Information is defined as

$$NMI(U,V) = \frac{MI(U,V)}{\sqrt{MI(U,U)MI(V,V)}}$$

**Adjusted Rand Index** (Hubert & Arabie, 1985; Skl) Suppose R is the groundtruth clustering and S is a partition, the RI of S is given as follows. Let $a$ be the number of pairs of elements that are in the same set in R as well as in S; $b$ be the number of pairs of elements that are in diferent sets in R, and different sets in S. Then

$$RI = \frac{a+b}{\binom{n}{2}}$$

$$ARI = \frac{RI - \mathbb{E}[RI]}{\max(RI) - \mathbb{E}[RI]}$$

## A.5 MORE DATASETS RESULTS

In Table 7, we show evaluation metrics on other four datasets for our method on both train and test split.

Table 7: More datasets: Clustering evaluation metrics of ConCURL on train and test splits

| Datasets | Intel | | | Caltech101 | | | CUB | | | AwA2 | | |
|---|---|---|---|---|---|---|---|---|---|---|---|---|
| Split\Metrics | Acc | NMI | ARI | Acc | NMI | ARI | Acc | NMI | ARI | Acc | NMI | ARI |
| Train | 0.910 | 0.801 | 0.800 | 0.339 | 0.651 | 0.223 | 0.127 | 0.452 | 0.033 | 0.539 | 0.681 | 0.448 |
| Test | 0.899 | 0.783 | 0.776 | 0.361 | 0.691 | 0.218 | 0.113 | 0.442 | 0.025 | 0.542 | 0.685 | 0.446 |

## A.6 ABLATION STUDY ON THE LOSSES

In this subsection, we study the effect of weights $\alpha, \beta$ and $\gamma$ on the final metrics. Results for the weight configuration corresponding to $\alpha = 1, \beta = 1, \gamma = 1$ is what is shown in the main paper. For the case of ($\alpha = 1, \beta = 0, \gamma = 0$), we computed the cluster accuracy, NMI, ARI by computing the embeddings of all the data output by the representation learning algorithm used for $L_1$ (here Grill et al. (2020)). Then we computed a K-means clustering on the embeddings (the target projection layer embeddings in this case) to obtain a partition of the data, and follow the same procedure mentioned in A.4. For all the experiments in this section, we trained only on the train split of the datasets.

Table 8: Cluster metrics evaluation on data points that available during training

| Datasets | ImageNet-10 | | | ImageNet-Dogs | | | STL10 | | | Intel | | | Caltech101 | | | CUB | | | AwA2 | | |
|---|---|---|---|---|---|---|---|---|---|---|---|---|---|---|---|---|---|---|---|---|---|
| Methods\Metrics | Acc | NMI | ARI | Acc | NMI | ARI | Acc | NMI | ARI | Acc | NMI | ARI | Acc | NMI | ARI | Acc | NMI | ARI | Acc | NMI | ARI |
| $\alpha=1, \beta=0, \gamma=0$ | 0.818 | 0.843 | 0.757 | 0.492 | 0.464 | 0.289 | 0.29 | 0.31 | 0.137 | 0.889 | 0.764 | 0.758 | 0.348 | 0.641 | 0.212 | 0.134 | 0.460 | 0.041 | 0.528 | 0.713 | 0.416 |
| $\alpha=1, \beta=1, \gamma=0$ | 0.905 | 0.875 | 0.841 | 0.400 | 0.386 | 0.245 | 0.565 | 0.503 | 0.373 | 0.907 | 0.797 | 0.795 | 0.309 | 0.634 | 0.190 | 0.128 | 0.454 | 0.034 | 0.539 | 0.684 | 0.461 |
| $\alpha=1, \beta=1, \gamma=1$ | 0.922 | 0.877 | 0.852 | 0.452 | 0.447 | 0.288 | 0.623 | 0.514 | 0.428 | 0.910 | 0.801 | 0.800 | 0.339 | 0.651 | 0.223 | 0.127 | 0.452 | 0.033 | 0.539 | 0.681 | 0.448 |

Table 9: Cluster metrics evaluation on data points that were not available during training

| Datasets | ImageNet-10 | | | ImageNet-Dogs | | | STL10 | | | Intel | | | Caltech101 | | | CUB | | | AwA2 | | |
|---|---|---|---|---|---|---|---|---|---|---|---|---|---|---|---|---|---|---|---|---|---|
| Methods\Metrics | Acc | NMI | ARI | Acc | NMI | ARI | Acc | NMI | ARI | Acc | NMI | ARI | Acc | NMI | ARI | Acc | NMI | ARI | Acc | NMI | ARI |
| $\alpha=1, \beta=0, \gamma=0$ | 0.782 | 0.778 | 0.630 | 0.444 | 0.507 | 0.279 | 0.276 | 0.299 | 0.130 | 0.893 | 0.772 | 0.765 | 0.325 | 0.644 | 0.136 | 0.128 | 0.453 | 0.034 | 0.552 | 0.701 | 0.413 |
| $\alpha=1, \beta=1, \gamma=0$ | 0.884 | 0.867 | 0.811 | 0.408 | 0.441 | 0.238 | 0.560 | 0.484 | 0.362 | 0.903 | 0.787 | 0.785 | 0.325 | 0.667 | 0.181 | 0.117 | 0.445 | 0.027 | 0.541 | 0.681 | 0.456 |
| $\alpha=1, \beta=1, \gamma=1$ | 0.864 | 0.840 | 0.770 | 0.455 | 0.477 | 0.274 | 0.611 | 0.498 | 0.410 | 0.899 | 0.783 | 0.776 | 0.361 | 0.691 | 0.218 | 0.113 | 0.442 | 0.025 | 0.542 | 0.685 | 0.446 |

## A.7 T-SNE PLOTS

In figure 3, we show t-sne plot of the ImageNet-10 embeddings obtained from ConCURL trained model. One can clearly see the separation between various clusters with the exception of airline and airship clusters. Airline and airship clusters are mixed together on leftmost and righmost part of the t-sne plot.

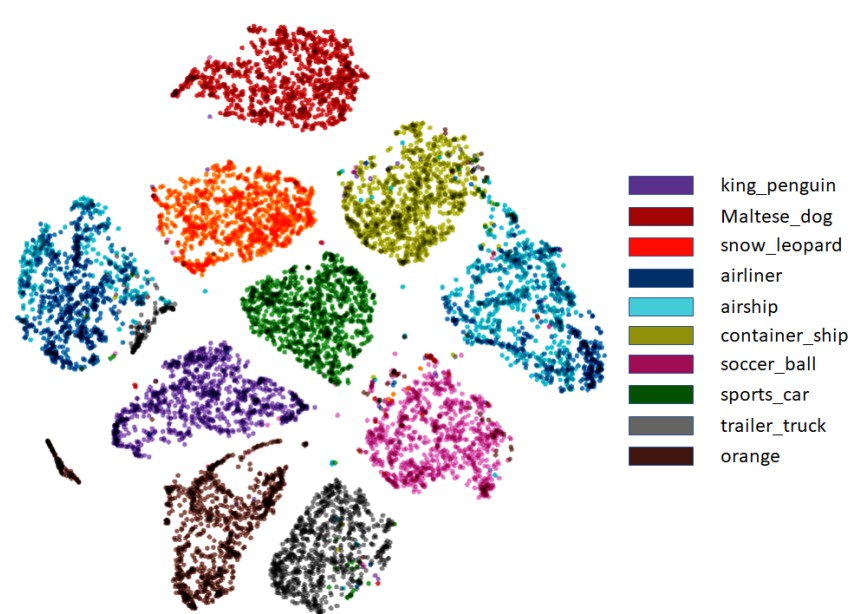

Figure 3: t-sne plot of ImageNet-10

