# OpenReview forum: "Consensus Clustering with Unsupervised Representation Learning"
_ICLR.cc/2021/Conference — Reject_

### Official Review · AnonReviewer3 · 2020-10-27
**Valid, but a very similar prior work ignored by the authors**

**Rating:** 5
**Confidence:** 4

**Review:**

The authors propose a learning-based approach for image clustering. In particular, similarly to recent algorithms fro unsupervised representation learning, such a DeepCluster, they propose to iterate between clustering the images in the feature space of the network and updating the network weights to respect the clusters. Two main differences with respect to DeepCluster-like algorithms is that they target the task of clustering itself, and do not evaluate the generalizability of the leaned representations for other tasks, and that they propose to use cluster ensembles to improve training robustness. In particular, they generate 2 clusterings of the images at every iteration by applying different sets of data augmentations and feature transformations to the input images. The objective is then not only to respect these clusterings but also to enforce consistency between them over time, thus improving representation invariance to irrelevant image details. In an experimental evaluation on a set of standard image clustering benchmarks they outperform prior work in most scenarios.

The idea is reasonable and the method seems sound, however a similar approach has been proposed in Zhuang et al., ICCV'19 (Local Aggregation for Unsupervised Learning of Visual Embeddings). In contrast to this work, the authors of Zhuang et al., used different runs of the clustering algorithm to obtain diverse clusterings, instead of transforming the images, but the overall approach is very similar. The authors seem to be not aware of that work.

In the current from it is not clear whether the proposed approach has any advantages over Zhuang et al. In the rebuttal the authors need to provide a discussion of their novelty with respect to that method as well as an experimental comparison on ImageNet.


I appreciate the authors' efforts to fairly compare to Zhuang et al. in the rebuttal, and I do find the preliminary evidence sufficient to establish that their approach outperforms LA on clustering metrics. However, the authors seem to miss the the point that LA is not just another consensus clustering approach which they forgot to include into literature review since it does not report clustering metrics. The main contribution of their work is combining consensus clustering with representation learning, which is exactly what the authors of LA had done before. It does seems that the particular approach proposed in this paper results in a better clustering performance, so the submission contains a valid contribution, but the relationship between the two methods needs to be discussed in a lot more detail, and they have to be throughly compared experimentally. I encourage the authors to improve the manuscript in this direction and resubmit to a different venue.

---

> ### Author Response · Authors · 2020-11-17
> **Response to Reviewer 3**
>
> We thank Reviewer 3 for the time and the feedback.
>
> **1:** "In the current form it is not clear whether the proposed approach has any advantages over Zhuang et al. In the rebuttal the authors need to provide a discussion of their novelty with respect to that method as well as an experimental comparison on ImageNet."
>
> **A:** We thank the reviewer for bringing Zhuang et al., ICCV’19 to our notice. It is unfortunate that we did not come across this paper.  We added this discussion to the revised document.
>
> Our main contribution is to formulate a consensus function using feature space transformations. The current form of the algorithm (and the consensus function) were chosen based on the soft clustering block used, and it can be easily adapted to other soft clustering approaches (such as IIC [B]). Whereas, the main contribution of LA is a novel local aggregation objective (which specifically follows from IR tasks) which uses a closest neighbors set generated through multiple runs of K-means clustering.
>
> Differences:
> Firstly, [A] does not show any clustering metric performance. It is not clear if [A] outperforms the other baselines discussed in our paper if evaluated on clustering metrics, as [A] uses labelled data to do evaluation (linear evaluation).
>
> [A] builds a novel Local Aggregation method on non-parametric instance discrimination tasks. These techniques usually require a large memory bank which is memory intensive. Our algorithm doesn't require memory intensive techniques like memory banks. During training, at each iteration [A] requires a nearest neighbor computation. Our method requires running a fast version of the iterative Sinkhorn-Knopp algorithm at each step [C].
>
> The way in which we generate the ensemble allows us to control and measure the diversity of the ensemble which can be useful in making algorithm design choices. Although [A] controls the ensemble by varying $H$ (number of clusterings) and $m$ (number of clusters in each clustering) which aptly suits the objective they are solving, their ensembles are limited to use k-means clustering. The authors show that other clustering approaches are either not scalable, or are not optimal. In our case, by applying feature space transformations, we have much more freedom in generating the ensemble. We used random projections, diagonal transformations, but there could be other transformations on the feature space.
>
> **REFERENCES**
>
> [A]  Zhuang, Chengxu, Alex Lin Zhai, and Daniel Yamins. "Local aggregation for unsupervised learning of visual embeddings." Proceedings of the IEEE International Conference on Computer Vision. 2019.
>
> [B] Ji, Xu, João F. Henriques, and Andrea Vedaldi. "Invariant information clustering for unsupervised image classification and segmentation." Proceedings of the IEEE International Conference on Computer Vision. 2019.
>
> [C] Caron, Mathilde, et al. "Unsupervised learning of visual features by contrasting cluster assignments." Advances in Neural Information Processing Systems 33 (2020).

---

> > ### Comment · AnonReviewer3 · 2020-11-18
> > **Reply**
> >
> > I thank the authors for their detailed response and discussion of the relationship of their method with Zhuang et al. However, mere claims that the proposed approach is more efficient and versatile are not sufficient for me to make a judgment. Please evaluate your method on ImageNet using the metrics in Zhuang et al., and use their released code to evaluate on the datasets used in your paper with corresponding metrics. Moreover, please report running time and memory requirements of both approaches.
> >
> > For versatility, please provide concrete examples of scenarios in which the approach of Zhuang et al. would not be applicable , but yours would. Ideally provide an evaluation in at least one such scenario.

---

> > > ### Author Response · Authors · 2020-11-24
> > > **Comparison with Zhuang et al., 2019**
> > >
> > > We thank the reviewer for the comment. As suggested, we evaluated Local Aggregation (LA) (Zhuang et al., 2019) on ImageNet-10, ImageNet-Dogs and compared the cluster metrics (Accuracy, NMI, ARI). We present the results in this table, and we explain the procedure.
> > >
> > > ImageNet-10:
> > >
> > > #ResNet-------------------| Acc    |    NMI   |    ARI    |\
> > > LA ResNet18-------------| 0.336  |   0.282  |   0.161 |\
> > > LA ResNet34-------------| 0.393  |   0.346  |   0.213 |\
> > > ConCURL-ResNet18---| 0.869   |    0.822 |  0.767 |\
> > > ConCURL-ResNet34---| **0.922** | **0.877** |  **0.852** |
> > >
> > >
> > > ImageNet-Dogs:
> > >
> > > #ResNet---------------------| Acc        NMI       ARI    |\
> > > LA ResNet18---------------| 0.201     0.157     0.063 |\
> > > LA ResNet34---------------| 0.201     0.133     0.06   |\
> > > ConCURL-ResNet18-----| 0.356     0.377     0.207 |\
> > > ConCURL-ResNet34-----| **0.452**     **0.447**    **0.288** |
> > >
> > > **Note**:
> > >
> > > - Due to time and resource constraints, we are not able to provide results of our algorithm on full ImageNet for clustering metrics (metric used in our paper), and linear evaluation (metric used in Zhuang et al., 2019).
> > >
> > > - However, since we use BYOL as the backbone in our method, we compare the cluster metrics and Linear evaluation metrics (on ImageNet) of BYOL with Zhuang et al., 2019. If time permits, we will upload these results before the deadline.
> > >
> > > - Note that the main aim of our paper is to get better performance on clustering and not linear evaluation (Zhuang et al., 2019 does not provide any clustering performance and is not compared in any state-of-the-art clustering papers. It is however a paper for linear evaluation and does not perform better than our proposed method for clustering according the results shown here)
> > >
> > > **Experiment Setup**:
> > >
> > > - **Code**: We used the Pytorch Implementation that has been validated by the first author of the paper.
> > >
> > > - **Training**: We trained the model for 500 epochs
> > >
> > > - **Hyper-parameter search**: We took the config file from the official PyTorch implementation (https://github.com/neuroailab/LocalAggregation-Pytorch/tree/master/config). We created 36 configuration files by varying learning rate and kmeans ‘k’. In the original config file of authors, they had used kmeans_k = 30000 (for 1.28 million images) . We scale the kmeans_k for ImageNet-10 (13000 images) and ImageNet-Dggs (19500 images) accordingly and try six different values.
> > >
> > > learning_rate_grid = [0.003, 0.005, 0.01, 0.03, 0.05, 0.1]
> > >
> > > kmeans_k_grid = [10, 15, 100, 310, 452, 500] # kmeans k
> > >
> > > The background neighbors are 4096 as used in the paper. We will provide configuration files in the supplementary material. We ran ResNet18 experiments for the full hyperparameter search (36 experiments) and evaluated the cluster metrics. Additionally, we chose the top 5 choices of hyperparameters from above, and also ran ResNet34 experiments with those parameters, for both the datasets.
> > >
> > > Since we report ConCURL results on ResNet34, we additionally trained ConCURL on ResNet18 for a fair comparison with LA results (these are not necessarily the best results with ResNet18 since we did not perform a hyper-parameter search for ConCURL)
> > >
> > > - **Evaluation**: The code repository uses a different version of ResNet (PreActResNet). We take the output of the layer before the final dense layer. For ResNet18, the output dimensions of this layer is (512,7,7) and we take the mean along (1,2) dimensions and use the resulting 512 dimension vector (for ResNet34 it is 2048). We compute k-means clustering on these embeddings using Faiss for the train split of the data, and compute the cluster metrics as mentioned in our paper.
> > >
> > >
> > > [A] Zhuang, Chengxu, Alex Lin Zhai, and Daniel Yamins. "Local aggregation for unsupervised learning of visual embeddings." Proceedings of the IEEE International Conference on Computer Vision. 2019.

---

> > > > ### Comment · AnonReviewer3 · 2020-11-25
> > > > **Response**
> > > >
> > > > I thank the authors for a thorough attempt to address my concerns. The preliminary results do indicate that the proposed approach has advantage over Zhuang et al. when evaluated for clustering. However, the gap in performance is really large, which requires a clear explanation given that the two approaches are fairly similar. So far, the authors did not even clearly acknowledge in the manuscript that the method of Zhuang et al. uses similar consensus clustering ideas.
> > > >
> > > > The argument that evaluating their method on the full ImageNet is computationally expensive is not particularly convincing either. Training LA on ImageNet takes about a day, and the authors claim that their approach is more efficient than LA, so I don't see how two weeks were not sufficient to provide an evaluation.
> > > >
> > > > Considering this and the other reviews, I intend to keep my original score, unless other reviewers are convinced that the paper has merit in its current form. This is a promising work, but a more thorough experimental evaluation and comparison to prior literature is required to reach the publication bar.

---

> > > > > ### Author Response · Authors · 2020-11-25
> > > > > **Response to Reviewer 3**
> > > > >
> > > > > We respect the views of the reviewer but we want to make some corrections to reviewers’s points:
> > > > >
> > > > > - “Training ImageNet in one day”: We believe that we addressed all concerns that were materialistically possible in the duration of the reviews. Reviewer is not right to say that one can train LA (Zhuang et al.) on imagenet in 1 day. For authors of LA, training took close to one week on two GPUs (Please look at https://github.com/neuroailab/LocalAggregation-Pytorch/issues/2).
> > > > >
> > > > > - “Literature of consensus clustering”: We do include LA (Zhuang et al.)  in our Literature review and we thank the reviewer for pointing this out. We also mention that their objective is based on robust clustering with multiple runs of k-means in the paper. We agree that LA uses ideas from consensus clustering approach.  But  LA (Zhuang et al.)  itself  does not mention a single paper in the consensus clustering in it’s literature (please look at our paper for the rich history of consensus clustering). We could have come across LA (Zhuang et al.)  if it had either mentioned consensus clustering papers or given clustering accuracy metric on standard datasets used in clustering with representation learning literature.
> > > > >
> > > > > - None of the state-of-the-art clustering work compares against LA (Zhuang et al.). We found it unfair when a reviewer asked us to compare against LA (Zhuang et al.). But we still have done hyperparameter search for LA (Zhuang et al.) and presented results (please look at our last comment).
> > > > >
> > > > > - As mentioned in our previous response, we report clustering metrics of LA on ImageNet (using pretrained weights from the official tensorflow implementation), and compare with the clustering metrics of BYOL (which we used as a backbone) using pretrained weights released on the official implementation.
> > > > >
> > > > > #ResNet-------------------| Acc     |    NMI    |    ARI    |\
> > > > > LA ResNet18-------------| 0.12    |   0.382  |   0.04    |\
> > > > > BYOL ResNet50---------| 0.319  |   0.580  |   0.179  |
> > > > >
> > > > > - Moreover note that none of the state-of-the-art clustering papers use full ImageNet data [B,C,D,E]. Also, linear evaluation results from [A] show that BYOL outperforms LA. But note that linear evaluation is not the purpose of the paper and our focus is on clustering with representation learning.
> > > > >
> > > > > We agree with the reviewer that LA (Zhuang et al.) does use ideas from consensus clustering. But we do not find it fair that we were asked to compare against a method that does not present any clustering performance metric (like Acc, NMI and ARI). Despite that, we presented empirical evidence that our method does perform better. It is unfortunate that the reviewer is not ready to change mind even after giving strong empirical evidence.
> > > > >
> > > > >
> > > > > [A] Grill, Jean-Bastien, et al. "Bootstrap Your Own Latent-A New Approach to Self-Supervised Learning." Advances in Neural Information Processing Systems 33 (2020).
> > > > >
> > > > > [B] Huang, Jiabo, Shaogang Gong, and Xiatian Zhu. "Deep Semantic Clustering by Partition Confidence Maximisation." In Proceedings of the IEEE/CVF Conference on Computer Vision and Pattern Recognition, pp.
> > > > > 8849-8858. 2020.
> > > > >
> > > > > [C] Wu, Jianlong, Keyu Long, Fei Wang, Chen Qian, Cheng Li, Zhouchen Lin, and Hongbin Zha. "Deep comprehensive correlation mining for image clustering." In Proceedings of the IEEE International Conference on Computer Vision, pp. 8150-8159. 2019.
> > > > >
> > > > > [D] Ji, Xu, Joao F. Henriques, and Andrea Vedaldi. "Invariant information distillation for unsupervised image segmentation and clustering." arXiv preprint arXiv:1807.06653 2, no. 3 (2018): 8.
> > > > >
> > > > > [E] Chang, Jianlong, Yiwen Guo, Lingfeng Wang, Gaofeng Meng, Shiming Xiang, and Chunhong Pan. "Deep Discriminative Clustering Analysis." arXiv preprint arXiv:1905.01681 (2019).

---

### Official Review · AnonReviewer2 · 2020-10-28
**A good attempt at combining ensemble learning approaches with deep clustering**

**Rating:** 5
**Confidence:** 3

**Review:**

This paper studies the effect of combining ensemble learning approaches with deep clustering. The paper wants to show that ensemble learning methods, in particular consensus clustering, can improve the clustering accuracy when combined with general representation learning/clustering blocks. However, I am not sure that the results presented in the paper are enough to support the claims.

The paper's  pros are:
(+) It Is the first to combine ensemble methods with deep clustering models. Although ensemble methods have been widely applied, studying consensus clustering in the current problem setting is novel.
(+) It Is the first to be able to have an ensemble deep clustering algorithm that gains empirically over other state-of-the-art models, showing the ideas to be potentially effective.
(+) The writing is in general clear and undestandable.

The paper's cons are:
(-) The wording in the abstract is a bit confusing in the sense that after reading it one might think the algorithm does consensus clustering first and uses the clustering to learn better representations of the input data. Although this is clarified later in the main body.
(-) The description of the main algorithm seems to be more intuitive than innovative. Some algorimic design choices are not very convincing. For example, the choice of using random projections on embedding to produce different clusterings, although an interesting idea, makes me wonder why it is necessarily a good way to introduce randomness into the whole framework. The authors can expand their discussion on this. I also remain dubious about why different representation instead of different clustering methods
Also, since the authors meant the idea to be applicable to general representation learning/deep clustering blocks, I'm not sure the current experimental data in the paper can lead to that conclusion.
(-) Using the performance metrics provided by the authors, I find it a bit hard to conclude that the proposed algorithm has a significant advantage over state of the art methods, especially PICA. Also, the fluctuation in performance metrics caused by different parameter settings seems to be, in magnitude, at least comparable to the margin of ConCURL over the baselines.

Overall, I think this paper contains  interesting seed ideas such as combining consensus clustering with representation learning and making use of the learned representation to generate multiple clusterings. These seed ideas could be good for this venue. However, the work is still premature and flawed by crude algorithm/experiment design.  The quality can be significantly improved if the authors can give a more general algorithmic framework (since the authors meant the ideas to be applicable to general representation learning/clustering algorithms), equipped with more thorough experimental investigation to support the applicability and superiority of the current approach.

---

> ### Author Response · Authors · 2020-11-17
> **Response to Reviewer 2: Part 1/2**
>
> We thank Reviewer 2 for the time and the feedback.
>
> **1:** “For example, the choice of using random projections on embedding to produce different clusterings, although an interesting idea, makes me wonder why it is necessarily a good way to introduce randomness into the whole framework. The authors can expand their discussion on this. I also remain dubious about why different representation instead of different clustering methods”
>
> **A:** We thank the reviewer for raising this interesting point. Indeed, there are many ways to introduce randomness into the framework. The next paragraph will explain why we preferred to use feature transformations as compared to multiple algorithms.  Then, we discuss why we use Random Projections/Diagonal transformations on the representations
> Ideally, it is possible to use different clustering methods to generate the ensemble and it was pursued in the Consensus Clustering literature. However, since we are learning representations simultaneously using backpropagation, it is natural to use a clustering algorithm in which we can update its parameters via backpropagation. It is in-fact possible to use clustering algorithms like IIC that enable parameter updates, however, if multiple such algorithms are used to generate the ensemble, the backward computation graph consumes an enormous amount of memory. For large datasets (and large architectures), such methods may not be scalable, however by using feature transformations, the amount of such memory overhead is small. By fixing a stable clustering algorithm, we can generate arbitrarily large ensembles by applying different transformations on the representations, and we can also control the diversity of the ensembles by trying several transformations.
>
> Random projections were successfully used in Consensus Clustering previously [A]. By generating ensembles using random projections, we have control over the amount of diversity we can induce into the framework, by varying the dimension of the random projection. We measure the diversity of the ensemble by performing the following experiment. For each component of the ensemble, we use the softmax probability estimates $\mathbf{p}$ and compute cluster assignments by taking an $\arg\max$ on $\mathbf{p}$ of each image. If there are M components in the ensemble, we get M such cluster assignments. We then compute a pairwise NMI (Normalized Mutual Information) (similar analysis to [A]) between every two components of the ensemble, and compute the average and standard deviation of the pairwise NMI across the $M*(M-1)/ 2$ pairs. We observe from Figure 2 that the pairwise NMI increases as training progresses and becomes closer to 1. At the beginning of training, the ensemble is very diverse (small NMI score with a larger standard deviation); and as training progresses, the diversity is reduced (large NMI score with a smaller standard deviation).
>
> In addition to Random Projections, we also used diagonal transformations [B] where different components of the representation vector are scaled differently. [B] illustrate that such scaling enables a diverse set of clusterings which is helpful for their meta clustering task.
>
> **2:** “Also, since the authors meant the idea to be applicable to general representation learning/deep clustering blocks, I'm not sure the current experimental data in the paper can lead to that conclusion”
>
> **A:** The motivation of this paper is to highlight that ensembles can be useful in deep clustering/representation learning. Therefore, our algorithm choices were based on some recent state-of-the-art results for unsupervised learning. However, due to the limited time/computation, we could not extend our analyses to other general representation learning/deep clustering blocks, but this is certainly a future direction we want to pursue.
>
>
> **REFERENCES**
>
> [A] Fern, Xiaoli Z., and Carla E. Brodley. "Random projection for high dimensional data clustering: A cluster ensemble approach." Proceedings of the 20th international conference on machine learning (ICML-03). 2003.
>
> [B] Hsu, Kyle, Sergey Levine, and Chelsea Finn. "Unsupervised learning via meta-learning." arXiv preprint arXiv:1810.02334 (2018).

---

> ### Author Response · Authors · 2020-11-17
> **Response to Reviewer 2: Part 2/2**
>
>
> **3:** “Using the performance metrics provided by the authors, I find it a bit hard to conclude that the proposed algorithm has a significant advantage over state of the art methods, especially PICA.”
>
> **A:** We used the popular Iman and Daveport Test to calculate the p-value for statistical significance test [C]. Our proposed method ConCURL was consistently ranked at the top for all three metrics ACC, ARI and NMI with p-value of $1.34 \times 10^{-4}$ , $4.57 \times 10^{-5}$ and $0.00145$ respectively. This shows that results presented in the paper are statistically significant.
>
>
> **4:** “Also, the fluctuation in performance metrics caused by different parameter settings seems to be, in magnitude, at least comparable to the margin of ConCURL over the baselines.”
>
> **A:** We updated our results section to show a fair comparison of our method against PICA. We trained our algorithm to use ‘train+test’ split during training, and compare it with the results reported in PICA on STL10 (Note: For STL10, PICA results use ‘train+test+unlabelled’ where the unlabeled set is 100k images. However, we outperform PICA without using unlabelled split during training). We perform similar analysis for CIFAR10, and CIFAR100. The results are shown in the table below.
>                         |           STL10                |             CIFAR10          |            CIFAR100         |
>                         |   Acc     NMI     ARI    |   Acc      NMI      ARI   |   Acc      NMI      ARI   |
> --------------------------------------------------------------------------------------------------------------|
> PICA               |  0.713  0.611  0.531    |  0.696  0.591  0.512   |  0.337  0.310  0.171   |
> ConCURL       |  0.728  0.634  0.554    |  0.720  0.608  0.520   |  0.385  0.377  0.223   |
>
> We also reproduced PICA results (https://github.com/Raymond-sci/PICA) and modified the parameters to use only ‘train’ split for training, and compare with the results of our algorithm when trained only on the ‘train’ split. The table below shows train accuracy for the above setup. From these results, it is evident that ConCurl outperforms PICA in all the cases.
>                         |              STL10              |              CIFAR10           |              CIFAR100        |
>                         |      Acc     NMI      ARI  |      Acc     NMI      ARI  |      Acc     NMI      ARI  |
> -----------------------------------------------------------------------------------------------------------------|
> PICA               |  0.494    0.424   0.297 |  0.559    0.510    0.402 |  0.295    0.273	 0.138  |
> ConCURL       |  0.623    0.514   0.428 |  0.705    0.545    0.507 |  0.366    0.351    0.195  |
>
> We also conducted ablation studies to examine the performance of the algorithm under various choices of the hyperparameters, and analyse the results.  The results for these ablation studies are presented below for STL10 dataset (when trained and evaluated on the ‘train’ split). The results fluctuate with a margin of +- 0.06 and still outperform the other baselines (PICA: Acc: 0.494, NMI: 0.424, ARI: 0.297).
> In the following table, we used diagonal transformations, and varied the number of transformations.
> #transformations| Acc  	  NMI  	 ARI    |
> ---------------------------------------------------------|
>                          10   | 0.567     0.514     0.373 |
>                          30   | 0.534     0.481     0.338 |
>                          50   | 0.582     0.525     0.397 |
>                         100  | 0.530     0.489     0.344 |
>
> The following table contains results with a fixed dimension of random projection (=64), and varies the number of transformations.
> #transformations	| Acc  	 NMI  	ARI	|
> -----------------------------------------------------------|
>                           10   | 0.520	 0.468	0.317	|
>                           30   | 0.586	 0.515	0.386	|
>                           50   | 0.562	 0.512	0.377	|
>                           100 | 0.533	 0.503	0.343	|
>
> The following table contains results with a fixed number of transformations (=100), and varies the dimension of projection  (the original embedding size is 256).
>          Dim of RP         | Acc     NMI       ARI  |
> ---------------------------------------------------------|
>                             64  | 0.533    0.503    0.342 |
>                            128 | 0.487    0.488    0.310 |
>                            256 | 0.525    0.502    0.338 |
>                            512 | 0.584    0.499    0.375 |
>
>
> **REFERENCES**
>
> [C] Garcia, Salvador, and Francisco Herrera. "An extension on``statistical comparisons of classifiers over multiple data sets''for all pairwise comparisons." Journal of machine learning research 9, no. Dec (2008): 2677-2694.

---

### Official Review · AnonReviewer4 · 2020-10-29
**The authors propose a deep learning approach to clustering that combines learning representations and ensemble learning. The aim is to learn a unifying data representation by creating a consensus on multiple clustering outputs. The type of clustering being pursued is a standard centroid-based clustering.**

**Rating:** 4
**Confidence:** 5

**Review:**

- The idea of combining learning representations and ensemble clustering is interesting and possibly promising. I think this paper in its present form is not ready to be published though.

- Cited literature on consensus clustering is old and outdated (up to 2005) and does not capture the state of the art. More recent work exists that outperforms the ones cited in this paper. The authors should also consider comparing their method with the state of the art on consensus clustering which is not deep learning based.

- The description of the algorithm is often too vague and is fragmented in mini-sections, and the flow that brings them together is unclear. The individual pieces are not novel and no novel technical challenges or contributions are presented. Several steps and settings are ad-hoc and not well-motivated. What does it mean exactly: "Performing a forward pass on the two views results in feature vectors f1,f2."? How are f1 and f2 obtained? What's the motivation for using a multi-layer perceptron to further reduce f1 and f2? Why should we expect this to produce a good representation for clustering? How were the dimensionalities of the hidden and output layers chosen?

- The notation in Eq. (2) is not clearly defined. What is a transportation polytope?

- Writing and organization need major work. There are many vague statements that are not well-supported or motivated.

- How does the set of observations X defined in 3.1 relate with the input batch X_b in 3.1.2?

- The process of generating two views of the input images is not clearly defined. It seems that this process mainly consists in adding random noise to the images? This is quite different from the notion of different *views* of a data object (e.g. image) in the literature. Also, why 2 views and not 3 or more?

- I don't find Figure 1 particularly useful. Part (a) does not help at all in understanding the approach.

- What's the size of a typical ensemble being generated? The paper mentions M transformations. What is the typical M value. In the experiments M ranges from 10 to 100. How sensitive is the approach to the specific transformations being applied and to the number of transformations?

- The relationship and distinction between the views and the ensemble need to be better clarified.

- I find the explanation above Eq (4) unconvincing. How diverse are the probability estimations generated by each component of the ensemble? Have the authors performed any analysis on this?

- The authors should discuss assumptions and limitations of the proposed methodology. What assumptions does the proposed methodology make? Under which scenarios and data distributions is the methodology expected to perform well? What are its limitations?

- No insight or analysis on the results is provided. Often PICA outperforms the proposed method. I understand that PICA uses both training and test data, but since we are dealing with clustering and unsupervised data, I am not sure this is a real advantage for PICA.

---

> ### Author Response · Authors · 2020-11-17
> **Response to Reviewer 4: Part 1/3**
>
> We thank Reviewer4 for the time and the feedback.
>
> **1:** “Cited literature on consensus clustering is old and outdated (up to 2005)”
>
> **A:** We thank the reviewer for pointing this out. We have added recent papers in the survey.
>
>
> **2:** “The description of the algorithm is often too vague and is fragmented in mini-sections…”
>
> **A:** We modified the description of the algorithm to address the reviewer’s comments. Our motivation for this work is to show that consensus techniques can be effective in unsupervised learning. The main challenge in such consensus techniques is to build a consensus function.
>
>
> **3:** “What does it mean exactly: "Performing a forward pass on the two views results in feature vectors f1,f2."? How are f1 and f2 obtained? What's the motivation for using a multi-layer perceptron to further reduce f1 and f2? Why should we expect this to produce a good representation for clustering? How were the dimensionalities of the hidden and output layers chosen?”
>
> **A:** Suppose the two augmented views of a given image are denoted as $I^1$, $I^2$, passing these augmented views through the network gives us features $\mathbf{f}^1$, $\mathbf{f}^2$ of the two views. We use an MLP to further reduce these features motivated by the findings from SimCLR [A]. It was observed in [A] that using a non-linear projection layer improves the  representation quality before it. This technique was then also used in [B,C]. We choose the hidden layer and output dimension similar to that in [B], however since we used ResNet-34 (which has feature space dimension 512) as compared to ResNet-50 (which has feature space dimension 2048) in [B], we used a hidden layer size of 2048 instead of 4096. We have added this clarification in the paper.
>
>
> **4:** “The notation in Eq. (2) is not clearly defined. What is a transportation polytope?”
>
> **A:** We updated and defined the notation of eq. (2) in the document.
>
>
> **5:** “How does the set of observations X defined in 3.1 relate with the input batch X_b in 3.1.2?”
>
> **A:** We updated the document to make the distinction clear. Set of observations $\mathcal{X}$ refers to all the data available. The algorithm is a minibatch based algorithm, and the input batch $\mathcal{X}_b \subset \mathcal{X}$ represents the minibatch of size $B$ of the data used at a given iteration.
>
>
> **6:** “The process of generating two views of the input images is not clearly defined… Also, why 2 views and not 3 or more?”
>
> **A:** We thank the reviewer for pointing out the inconsistency. The different views aren’t the same as views in multi-view datasets [F]. The views referred to in this paper correspond to different augmented views that are generated by image augmentation techniques, such as ‘RandomHorizontalFlip’, ‘RandomCrop’ (see [A, B]). In our algorithm, we use 2 augmented views of each image in the loss for the sake of simplicity. Indeed it is possible to use more than 2 views. Authors of [C] propose MultiCrop: a way to use multiple views (2 global views of higher resolution, and 10 local views of lower resolution).
>
>
> **7:** “I don't find Figure 1 particularly useful. Part (a) does not help at all in understanding the approach.”
>
> **A:** We agree and we have moved the part (a) of the figure to appendix. We have kept part (b) of the figure in the main text and reference it in the main text for more clarity.

---

> ### Author Response · Authors · 2020-11-17
> **Response to Reviewer 4: Part 2/3**
>
> **8:** “What's the size of a typical ensemble being generated? The paper mentions M transformations. What is the typical M value. In the experiments M ranges from 10 to 100. How sensitive is the approach to the specific transformations being applied and to the number of transformations?”
>
> **A:** We added a section on ablation studies to examine the performance of the algorithm under various choices of the hyperparameters, and analyse the results.  The results for these ablation studies are presented below for STL10 dataset (when trained and evaluated on the ‘train’ split). M = 30,50 yield good results for the two types of transformations (random projections and diagonal transformations). We can also observe that there isn’t a large difference between results obtained using either of the transformations. The results fluctuate with a margin of +- 0.06 and still outperform the other baselines in almost all the cases (PICA:- Acc: 0.494, NMI: 0.424, ARI: 0.297).
>
> In the following table, we used diagonal transformations, and varied the number of transformations.
>
> #transformations| Acc  	  NMI  	 ARI    |\
> ---------------------------------------------------------|\
>                          10   | 0.567     0.514     0.373 |\
>                          30   | 0.534     0.481     0.338 |\
>                          50   | 0.582     0.525     0.397 |\
>                         100  | 0.530     0.489     0.344 |
>
> The following table contains results with a fixed dimension of random projection (=64), and varies the number of transformations.
>
> #transformations	| Acc  	 NMI  	ARI	|\
> -----------------------------------------------------------|\
>                           10   | 0.520	 0.468	0.317	|\
>                           30   | 0.586	 0.515	0.386	|\
>                           50   | 0.562	 0.512	0.377	|\
>                           100 | 0.533	 0.503	0.343	|
>
> The following table contains results with a fixed number of transformations (=100), and varies the dimension of projection  (the original embedding size is 256).
>          Dim of RP         | Acc     NMI       ARI  |\
> ---------------------------------------------------------|\
>                             64  | 0.533    0.503    0.342 |\
>                            128 | 0.487    0.488    0.310 |\
>                            256 | 0.525    0.502    0.338 |\
>                            512 | 0.584    0.499    0.375 |
>
>
>
> **9:** “The relationship and distinction between the views and the ensemble need to be better clarified.”
>
> **A:** For the purpose of this paper, we have 2 views. These views are generated as a part of a self-supervised task. Here 2 views come from any data augmentation of the image and if both views come from the same image, they should have the closer representations. Ensemble on the other hand is used after randomly projecting the representations to a new space and then predicting soft cluster assignments in this new space.
>
>
> **10:** “I find the explanation above Eq (4) unconvincing. How diverse are the probability estimations generated by each component of the ensemble? Have the authors performed any analysis on this?”
>
> **A:** There can be many clusterings for any given latent space (either by randomization introduced in the space or different clustering algorithms). We propose to use consensus clustering so that we can find a latent space that maximizes performance on many different clusterings. To make the method effective, we need more diversity in the clusterings among each component of the ensemble at the start and ideally the diversity should decrease with training and the clustering performance should get better. To confirm this, we measure the diversity of the ensemble by performing the following experiment. For each component of the ensemble, we use the softmax probability estimates $p$ and compute cluster assignments by taking an $\arg\max$ on $p$ of each image. If there are M components in the ensemble, we get M such cluster assignments. We then compute a pairwise NMI (Normalized Mutual Information) (similar analysis to [E]) between every two components of the ensemble, and compute the average and standard deviation of the pairwise NMI across the $M(M-1)/ 2$ pairs. We observe from Figure 2 that the pairwise NMI increases as training progresses and becomes closer to 1. At the beginning of training, the ensemble is very diverse (small NMI score with a larger standard deviation); and as training progresses, the diversity is reduced (large NMI score with a smaller standard deviation).

---

> ### Author Response · Authors · 2020-11-17
> **Response to Reviewer 4: Part 3/3**
>
>
> **11:** “The authors should discuss assumptions and limitations of the proposed methodology. What assumptions does the proposed methodology make? Under which scenarios and data distributions is the methodology expected to perform well? What are its limitations?”
>
> **A:** We thank the reviewer for this question and answering this question will definitely add more value to the paper. We have added the following discussion to the paper:
> First assumption of the paper is that “K” number of clusters are known. Second assumption is that the data sample is equally distributed among “K” clusters. Note that all state-of-the-art methods on datasets used make the same assumptions. These assumptions are important to make the methodology work because the final layer in our case needs to know the number of clusters so that we can evaluate it later for fair comparison with other methods. Also, the fast Sinkhorn-Knopp algorithm used to solve eq. (2) may not be optimal if the data sample is not equally distributed among  “K” clusters.
>
>
> **12:** “No insight or analysis on the results is provided. Often PICA outperforms the proposed method. I understand that PICA uses both training and test data, but since we are dealing with clustering and unsupervised data, I am not sure this is a real advantage for PICA”
>
> **A:** We added a section on ablation studies to examine the performance of the algorithm under various choices of the hyperparameters, and analyse the results. Please refer to Section 4.  We updated our results section to show a fair comparison of our method against PICA and outperform PICA on every dataset. We trained our algorithm to use ‘train+test’ split during training, and compare it with the results reported in PICA on STL10 (Note: For STL10, PICA results use ‘train+test+unlabelled’ where the unlabeled set is 100k images. However, we outperform PICA without using unlabelled split during training). We perform similar analysis for CIFAR10, and CIFAR100. The results are shown in the table below.
>                         |           STL10                |             CIFAR10          |            CIFAR100         |\
>                         |   Acc     NMI     ARI    |   Acc      NMI      ARI   |   Acc      NMI      ARI   |\
> --------------------------------------------------------------------------------------------------------------|\
> PICA               |  0.713  0.611  0.531    |  0.696  0.591  0.512   |  0.337  0.310  0.171   |\
> ConCURL       |  0.728  0.634  0.554    |  0.720  0.608  0.520   |  0.385  0.377  0.223   |
>
> We also reproduced PICA results (https://github.com/Raymond-sci/PICA) and modified the parameters to use only ‘train’ split for training, and compare with the results of our algorithm when trained only on the ‘train’ split. The table below shows train accuracy for the above setup. From these results, it is evident that ConCurl outperforms PICA in all the cases.
>                        |              STL10              |              CIFAR10           |              CIFAR100        |\
>                        |      Acc     NMI      ARI  |      Acc     NMI      ARI  |      Acc     NMI      ARI  |\
> -----------------------------------------------------------------------------------------------------------------|\
> PICA               |  0.494    0.424   0.297 |  0.559    0.510    0.402 |  0.295    0.273	 0.138  |\
> ConCURL       |  0.623    0.514   0.428 |  0.705    0.545    0.507 |  0.366    0.351    0.195  |
>
> **REFERENCES:**
>
> [A] Chen, Ting, et al. "A simple framework for contrastive learning of visual representations." arXiv preprint arXiv:2002.05709 (2020).
>
> [B] Grill, Jean-Bastien, et al. "Bootstrap Your Own Latent-A New Approach to Self-Supervised Learning." Advances in Neural Information Processing Systems 33 (2020).
>
> [C] Caron, Mathilde, et al. "Unsupervised learning of visual features by contrasting cluster assignments." Advances in Neural Information Processing Systems 33 (2020).
>
> [D] Garcia, Salvador, and Francisco Herrera. "An extension on``statistical comparisons of classifiers over multiple data sets''for all pairwise comparisons." Journal of machine learning research 9, no. Dec (2008): 2677-2694.
>
> [E] Fern, Xiaoli Z., and Carla E. Brodley. "Random projection for high dimensional data clustering: A cluster ensemble approach." Proceedings of the 20th international conference on machine learning (ICML-03). 2003.
>
> [F] Schops, Thomas, et al. "A multi-view stereo benchmark with high-resolution images and multi-camera videos." Proceedings of the IEEE Conference on Computer Vision and Pattern Recognition. 2017.

---

### Author Response · Authors · 2020-11-17
**Summary of changes**

We thank all the reviewers for the constructive feedback. We made an update to the paper with several changes based on reviewers' feedback and addressed any issues reviewers had.

1) In section 2, we have added more papers in the literature survey.
2) In section 3, we added more intuition of the algorithms and justification of choices we made. Especially we added section 3.2 describing why we chose "feature transformations" over "multiple clustering algorithms".
3) In section 4.1, we added results for our method that is trained on train+test on STL10, CIFAR10 and CIFAR100-20 (exactly like other baselines) to be consistent. We outperform all baselines including PICA. We have also added p-values for statistical significance test and results presented are statistically significant.
4) In section 4.2 and 4.3, we also added various ablation study results.

There are other minor changes that are included in the author's individual response. All the tables mentioned in the individual author response are also present in the paper.

---

### Decision · Program_Chairs · 2021-01-07
**Final Decision**

**Decision:**

Reject

**Comment:**

This paper proposes a model for learning using ensemble clustering. The reviewers found the general idea promising.
However, while promising, all reviewers noted that in its curent form the paper is not fit for publication. The reviewers pointed out missing references, issues with the abstract, lack of motivation for some of the algorithmic choices, limited novelty over clarity in the description of difference w.r.t. previous work.
Because of all these reasons, this paper does not meet the bar of acceptance. I recommend the authors take into account the feedback provided in the reviews and discussion and resubmit to another venue.